# SHMT: Self-supervised Hierarchical Makeup Transfer via Latent Diffusion Models

**Zhaoyang Sun**[1,3*]         **Shengwu Xiong**[1,2,5]         **Yaxiong Chen**[1]         **Fei Du**[3,4]

**Weihua Chen**[3,4]                **Fan Wang**[3,4]                **Yi Rong**[1,2†]

[1]School of Computer Science and Artificial Intelligence, Wuhan University of Technology
[2]Sanya Science and Education Innovation Park, Wuhan University of Technology
[3]DAMO Academy, Alibaba Group    [4]Hupan Laboratory    [5]Shanghai AI Laboratory

## Abstract

This paper studies the challenging task of makeup transfer, which aims to apply diverse makeup styles precisely and naturally to a given facial image. Due to the absence of paired data, current methods typically synthesize sub-optimal pseudo ground truths to guide the model training, resulting in low makeup fidelity. Additionally, different makeup styles generally have varying effects on the person face, but existing methods struggle to deal with this diversity. To address these issues, we propose a novel Self-supervised Hierarchical Makeup Transfer (SHMT) method via latent diffusion models. Following a "decoupling-and-reconstruction" paradigm, SHMT works in a self-supervised manner, freeing itself from the misguidance of imprecise pseudo-paired data. Furthermore, to accommodate a variety of makeup styles, hierarchical texture details are decomposed via a Laplacian pyramid and selectively introduced to the content representation. Finally, we design a novel Iterative Dual Alignment (IDA) module that dynamically adjusts the injection condition of the diffusion model, allowing the alignment errors caused by the domain gap between content and makeup representations to be corrected. Extensive quantitative and qualitative analyses demonstrate the effectiveness of our method. Our code is available at `https://github.com/Snowfallingplum/SHMT`.

## 1   Introduction

Recently, makeup transfer has become a popular application in social media and the virtual world. With its significant economic potential in e-commerce and entertainment, this technique is attracting widespread attention from the computer vision and artificial intelligence communities. Given a pair of source and reference face images, makeup transfer involves simultaneously focusing on the realism of the transferred result, the content preservation of the source image, and the makeup fidelity of the reference image. Although previous approaches [21, 3, 11, 4, 17, 24, 7, 29, 27, 16, 26, 41, 36, 47, 46] have made significant advances in image realism and content preservation , the challenge of achieving high-fidelity transfer of various makeup styles still remains unsolved.

The difficulties of makeup transfer mainly stem from two aspects. On the one hand, makeup transfer is essentially an unsupervised task, which means that there are no real transferred images that can be used as labeled targets for model training. To address this issue, previous methods typically synthesize a "pseudo" ground truth from each input source-reference image pair, as an alternative supervision

---

*Work done during internship of Zhaoyang Sun at DAMO Academy, Alibaba Group.
†Corresponding author: Yi Rong (yrong@whut.edu.cn).

38th Conference on Neural Information Processing Systems (NeurIPS 2024).

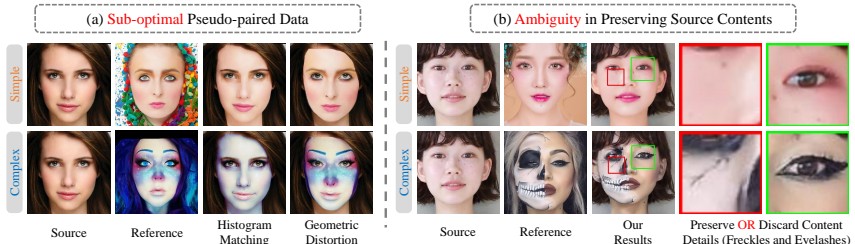

Figure 1: Illustration of two main difficulties in the makeup transfer task. (a) Due to the absence of paired data, previous methods utilize histogram matching or geometric distortion to synthesize sub-optimal pseudo-paired data, which inevitably misguide the model training. (b) Some source content details should be preserved in simple makeup styles but be removed in complex ones.

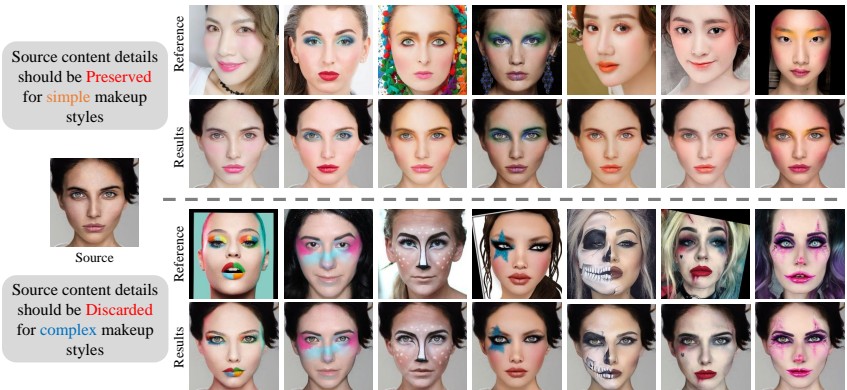

Figure 2: In addition to color matching, our approach allows flexible control to preserve or discard texture details for various makeup styles, without changing the facial shape.

signal. However, as shown in Figure 1(a), current pseudo-paired data synthesis techniques, including histogram matching [21, 29] and geometric distortion [11, 41, 36], fail to produce desirable outcomes. The reason is that histogram-matching-based methods often ignore the spatial properties of makeup styles, thus generating over-smoothed targets that lose most makeup details. And since the warping process in geometric-distortion-based methods relies solely on the shape information (e.g., facial landmarks) of input images, their pseudo targets usually contain undesired artifacts. Consequently, these sub-optimal pseudo-paired data will inevitably misguide the model training. This eventually results in the fact that most existing methods [21, 17, 24, 7, 41, 36, 47] generally exhibit low makeup fidelity, especially for those images with complex makeup details.

On the other hand, the diversity of different makeup styles can also lead to ambiguity in preserving source contents. In practice, makeup styles can range from natural, barely-there looks to elaborate and dramatic ones, each having a different impact on the person face. As shown in Figure 1(b), the content details of a source face (e.g., freckles and eyelashes) usually should be preserved in a simple makeup style, while they may be obscured in some complex ones due to the heavy use of cosmetics. An ideal model should be flexible enough to preserve or discard those source content details according to user preferences. However, all the previous approaches overlook this requirement.

To address the aforementioned dilemmas, we propose a novel Self-supervised Hierarchical Makeup Transfer (SHMT) method, which is built on the recent latent diffusion models [31]. For the first problem, considering the unsupervised nature of makeup transfer, we develop a self-supervised learning strategy following a "decoupling-and-reconstruction" paradigm. Specifically, given a face image, SHMT first extracts its content and makeup representations, and then simulates the makeup transfer procedure by reconstructing the original input from these decoupled information. In SHMT, the content representation of a face image includes its 3D shape and texture details, while the associated makeup representation is captured by destroying the content information from the input image through using random spatial transformations. In this way, SHMT works in a self-supervised manner, thus eliminating the misguidance of pseudo-paired data. To address the second issue, we

introduce a Laplacian pyramid [2] to hierarchically decompose the texture information in input image, allowing SHMT to flexibly control the preservation or discard of these content details for various makeup styles, as illustrated in Figure 2. Additionally, we propose an Iterative Dual Alignment (IDA) module in conjunction with the stepwise denoising property of diffusion models. In each denoising step, IDA utilizes the intermediate result to dynamically adjust the injection condition, which will help to correct the alignment errors caused by the domain gaps between the content and makeup representations. Our main contributions can be summarized as follows:

- We propose a novel makeup transfer method, named SHMT, which employs a self-supervised learning strategy for model training, thus getting rid of the misleading pseudo-pairing data adopted by previous methods.

- A Laplacian pyramid is introduced to hierarchically characterize the texture information, enabling SHMT to flexibly process these content details for various makeup styles.

- A new Iterative Dual Alignment (IDA) module is proposed, which dynamically adjusts the injection condition in each denoising step, such that the alignment errors caused by the domain gaps between the content and makeup representations can be corrected.

- Extensive qualitative and quantitative results indicate that SHMT outperforms other state-of-the-art makeup transfer methods. And additionally, the ablation studies demonstrate the robustness and the generalization ability of our SHMT method.

## 2   Related Works

### 2.1   Makeup Transfer

Over the past decade, makeup transfer [38, 12, 19] has gained increasing attention in the field of computer vision. BeautyGAN [21] designs a histogram matching loss and a dual input/output GAN [10] to simultaneously perform makeup transfer and removal. PairedCycleGAN [3] trains additional style discriminators to measure the local makeup similarity between the results and reference images. BeautyGlow [4] decomposes the latent vectors of face images derived from the Glow [18] framework into makeup and nonmakeup latent vectors. To address misaligned head poses and facial expressions, PSGAN [17, 24] utilizes an attention mechanism [40] to adaptively deform the makeup feature maps based on source images, while SCGAN [7] encodes component-wise makeup regions into spatially-invariant style codes. RamGAN [45] and SpMT [54] explore local attention to eliminate potential associations between different makeup components. Considering that histogram matching discards the spatial properties of the makeup styles, FAT [41] and SSAT [36, 37] design a pseudo-paired data synthesis based on geometric distortion. EleGANt [47] proposes a more effective pseudo-paired data by assigning varying weights to above two synthesis methods for performance improvement. For complex makeup styles, LAND [11] leverages multiple overlapping local makeup style discriminators to focus on the high-frequency makeup details. Additionally, CPM [29] applies a segmentation model to predict the mask of the makeup pattern. This segmented pattern is then pasted into semantically identical locations using UV [8] space.

Due to the unsupervised nature of makeup transfer, most of the above methods synthesize pseudo-paired data to guide model training. In this strategy, the quality of these pseudo-paired data is critical, leading many works [3, 41, 36, 29, 47] to strive for better synthesis techniques. With the help of the unprecedented generative capabilities of both GPT-4V and Stable Diffusion, a concurrent work Stable-Makeup [52] produces higher quality pseudo-paired data, thereby improving the performance of makeup transfer. Unlike these methods, our approach works in a self-supervised manner and eliminates the need for cumbersome pseudo-paired data synthesis.

### 2.2   Diffusion Models

Diffusion models generate realistic images through an iterative inverse denoising process. Recently, as competitors to generative adversarial networks [10], diffusion models have shown significant progress in numerous generative tasks, including Text-to-Image (T2I) generation [30, 33, 31], controllable editing [28, 51, 48], and subject-driven [9, 32, 5] or human-centric [22, 44, 43] synthesis. DDPM [15] proves the feasibility of recovering realistic images from random Gaussian noise. DDIM [35] enables fast and deterministic inference by transforming the sampling process into a non-Markovian

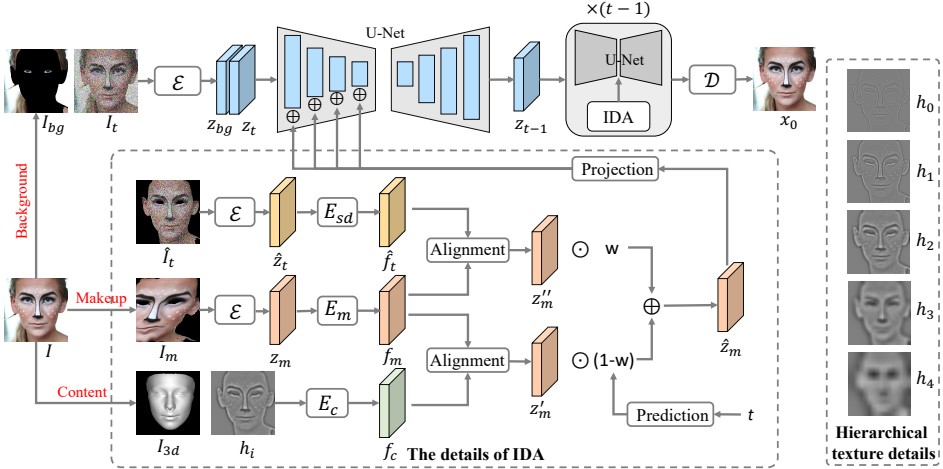

Figure 3: The framework of SHMT. A facial image $I$ is decomposed into background area $I_{bg}$, makeup representation $I_m$, and content representation $(I_{3d}, h_i)$. The makeup transfer procedure is simulated by reconstructing the original image from these components. Hierarchica texture details $h_i$ are constructed to respond to different makeup styles. In each denoising step $t$, IDA draws on the noisy intermediate result $\hat{I}_t$ to dynamically adjust the injection condition to correct alignment errors.

process. To reduce computational complexity while retaining high quality and flexibility, Latent Diffusion Models (LDM) [31] apply diffusion model training in the latent space of powerful pretrained autoencoders. With training on large datasets, both Imagen [33] and Stable Diffusio [31] elevate T2I synthesis to an unprecedented level.

Inspired by the above approach, we delve into the makeup transfer task based on diffusion models. Compared to previous GAN-based makeup transfer methods, our approach eliminates the need for adversarial training and tedious loss function design, while delivering enhanced performance.

## 3 Our Methodology

### 3.1 Preliminary

Our method is developed from Latent Diffusion Model (LDM) [31], which performs the diffusion process in the latent space to reduce the computational complexity of the model. In particular, LDM comprises three key components: an image encoder $\mathcal{E}$, a decoder $\mathcal{D}$, and an UNet denoiser $\epsilon_\theta$. Firstly, the encoder $\mathcal{E}$ compresses an image $I$ from the pixel space to a low-dimensional latent space $z_0 = \mathcal{E}(I)$, while the decoder $\mathcal{D}$ efforts to reconstruct the original image $I$ from the latent variable $z_0$ (e.g. $\mathcal{D}(z_0) = x_0 \approx I$). Then, a UNet denoiser $\epsilon_\theta$ is trained to predict the applied noise in the latent space. The optimization process can be defined as the following formulation:

$$\mathcal{L}_{ldm} = \mathbb{E}_{z_t, c, \epsilon \sim \mathcal{N}(0,1), t \sim \mathcal{U}(1,T)}[\| \epsilon_\theta(z_t, c, t) - \epsilon \|_2], \tag{1}$$

where $z_t$ is the noisy latent variable at the timestep $t$, $c$ is the conditioned signal, $\epsilon$ is randomly sampled from the standard Gaussian distribution, and $T$ is the defined maximum timestep.

During inference, $z_T$ is sampled from a random Gaussian distribution. The UNet denoiser $\epsilon_\theta$ then iteratively predicts the noise in the latent space at each timestep $t$ and restores $z_0$ via a sampling process (e.g. DDPM [15] or DDIM[35]). Finally, $z_0$ is reconstructed by the decoder $\mathcal{D}$ to obtain the generated image. See [31] for more details.

### 3.2 Overview

The goal of makeup transfer is to generate a new facial image that preserves the content information (e.g., background, facial structure, pose, expression) of a source image while applying the makeup style of a reference image. To achieve this, we propose SHMT, with its framework outlined in Figure

3. In SHMT, we craft a self-supervised strategy for model training in Section 3.3 and design an Iterative Dual Alignment (IDA) module to correct alignment errors in Section 3.4. The training and inference process of SHMT is presented in Section 3.5.

## 3.3 Self-supervised Strategy

Following a "decoupling-and-reconstruction" paradigm, we craft a self-supervised strategy for makeup transfer. The main idea is to separate content and makeup representations from a facial image, and then reconstruct the original image from these components.

**Foreground and background segmentation.** Makeup transfer is a localized modification task confined to the facial area. As such, a pre-trained face parsing model [49] is used to segment the foreground and background areas from the original image $I$. In makeup transfer, the background area $I_{bg}$ (which includes hair and clothes) is a known component. Hence, the background area $I_{bg}$ is input to the latent diffusion model along with the noisy image $I_t$. The goal of the model is to inpaint the unknown facial area, using subsequent content and makeup representations as conditions.

**Makeup Representation.** In our approach, the makeup representation is derived by destroying the image's content information. Specifically, we apply a sequence of spatial transformations to the foreground image $I_{fg}$. These transformations include random cropping, rotation, and elastic distortion to create variation. As illustrated in Figure 3, the distorted foreground image $I_m$, while losing the majority of the content information, retains the makeup information well. In addition, these transformations effectively mimic semantic misalignment scenarios and facilitate the robustness of the model to poses and expressions [17, 7].

**Content Representation.** Inspired by physical face modeling [12, 19, 20], the content representation of a face is simplified into two main components: face shape and texture details.

*Face Shape*: The face shape, which determines the facial structure, head pose, and expression, is a crucial component to preserve during makeup transfer. In this context, a typical 3D face reconstruction model, known as 3DDFA-V2 [13], is employed to extract the face shape $I_{3d}$ from the original image $I$. To match the resolution of the latent space of LDM, we perform a pixel unshuffle operation [34] to downsample $I_{3d}$.

*Texture Details*: Texture details are incorporated to complement the excessively smooth face shape. Considering the ambiguity of texture detail preservation, the Laplace pyramid [2] is introduced to build *hierarchical texture details*. To prevent color disturbances, the foreground image $I_{fg}$ is first converted to a grayscale image $\hat{I}_{fg}$ of $h \times w$ pixels. Then we downsample it by applying a fixed Gaussian kernel to produce a low-pass prediction $l_1 \in \mathbb{R}^{\frac{h}{2} \times \frac{w}{2}}$. The high-frequency component $h_0$, which is treated as texture details, can be calculated as $h_0 = \hat{I}_{fg} - \hat{l}_1$, where $\hat{l}_1$ is upsampled from $l_1$. By replacing $\hat{I}_{fg}$ with $l_1$ and repeating these operations, we obtain a series of hierarchical texture details $[h_0, h_1, \cdots, h_L]$, where $L$ refers to the decomposition level of the Laplace pyramid. These texture details, whose resolution gradually halves and ranges from fine to coarse, are illustrated in Figure 3. If the resolution of the texture details exceeds that of the latent space, we downsample it using a pixel unshuffle operation [34]. Otherwise, the bilinear interpolation is used for upsampling.

Note that only one texture detail $h_i$ is concatenated to the face shape $I_{3d}$ as a complete content representation. When $h_i$ is a fine texture detail, our model only needs to distill the low-frequency makeup information from the makeup representation to reconstruct the image. This is suitable for simple makeup styles. When $h_i$ is a coarse texture detail, our model must also distill the high-frequency makeup information from the makeup representation to ensure the recovery of the image. This is suitable for complex makeup styles.

## 3.4 Iterative Dual Alignment

To reconstruct the original image, spatial attention [40] is utilized to semantically align distorted makeup representation with content representation. At each timestep, we constructs a pixel-wise correlation matrix $M$ by calculating the cosine similarity as:

$$M(i, j) = \frac{f_c(i)^T f_m(j)}{\|f_c(i)\|_2 \|f_m(j)\|_2}, \tag{2}$$

where $f_c = E_c(I_{3d}, h_i)$, $f_m = E_m(\mathcal{E}(I_m))$ denote the semantic features extracted by encoders $E_c$ and $E_m$. $f(i)$ represents the feature vector of the $i$-th pixel in $f$ and $M(i, j)$ indicates the element at the $(i, j)$-th location of $M$. We consider the correlation matrix $M$ as a deformation mapping function, and use it to spatially deform the feature maps $z_m = \mathcal{E}(I_m)$ of makeup representations:

$$z'_m = \sum_j Softmax(M(i,j)/\tau) \cdot z_m, \tag{3}$$

where $Softmax(\cdot)$ denotes a softmax computation along the column dimension, which normalizes the element values in each row of $M$, and $\tau > 0$ is a temperature parameter. Theoretically, the deformed feature maps $z'_m$ is semantically aligned with the content representation. However, due to the domain gap between content and makeup representations, we find that alignment errors occur frequently in $z'_m$. Considering the property that noisy intermediate result $I_t$ is gradually moving closer to the real image domain (e.g., the makeup representation domain), we propose a Iterative Dual Alignment (IDA) module to address the above issue. At each timestep, it calculates an extra alignment prediction $z''_m$ between the noisy intermediate result $\hat{I}_t$ and makeup representation $I_m$ to dynamically correct the previous alignment prediction $z'_m$. $\hat{I}_t$ is the result of $I_t$ removing the background area. Since the noise degree of $\hat{I}_t$ is determined by timestep $t$, a MLP module predicts the percentage $w$ of two alignment predictions from timestep $t$:

$$\hat{z}_m = (1 - w)z'_m + wz''_m, \tag{4}$$

where $w = \text{MLP}(t)$ and the MLP module consists of two fully connected layers and ends with a Sigmoid activation layer. Finally, the mixed prediction $\hat{z}_m$ is concatenated with the content representation and injected into the denoiser's encoder through a projection module consisting of $1 \times 1$ convolution. In addition to correct alignment errors, IDA has two other advantages. First, because $\hat{z}_m$ is semantically aligned with the content representation, IDA can control the makeup style in a spatially-aware manner. Corresponding results are displayed in the supplementary material. Second, IDA is relatively lightweight, with only approximately 11M parameters.

### 3.5 Training and Inference

During training, the parameters of the pre-trained autoencoder are fixed, the U-net denoiser $\epsilon_\theta$ and IDA module are jointly optimized from scratch under the original objective function in Equation 1. At inference, the model receives the background area $I_{bg}$ and content representation ($I_{3d}$ and $h_i$) from the source image, as well as the undeformed makeup representation $I_m$ from the reference image, to generate the makeup transfer result.

## 4 Experiments

### 4.1 Experimental settings

**Datasets.** Following [21, 17], we randomly select 90% of the images from the MT dataset [21] as training samples and the rest as test samples. In addition, Wild-MT [17] and LADN [11] datasets are also used to validate the performance and generalization capability of our model. The images in the Wild-MT dataset contain large pose and expression variations, and the LADN dataset collects a number of images with complex makeup styles.

**Implementation Details.** In our experiments, we discover that the autoencoder with a downsampling factor of 4 preserves texture details better than the one with a factor of 8. Therefore, the autoencoder with a downsampling factor of 4 is selected, and the SHMT model is trained at a resolution of $256 \times 256$. The specific structure of the UNet denoiser $\epsilon_\theta$ remains the same as the LDM [31], with IDA module replacing the original conditional injection module. In Equation 3, $\tau$ is set to 100. We train the model with Adam optimizer, learning rate of 1e-6 and batch size of 16 on a single A100 GPU. Our model is trained for $250, 000$ steps in about 5 days. For sampling, we utilize 50 steps of the DDIM sampler [35].

**Evaluation Metrics.** In order to comprehensively and objectively compare the different methods, we choose three evaluation metrics, *FID*, *CLS* and *Key-sim*. Following [24], *FID* [14] is calculated between reference images and transferred results to indicate *image realism*. The lower the *FID*, the

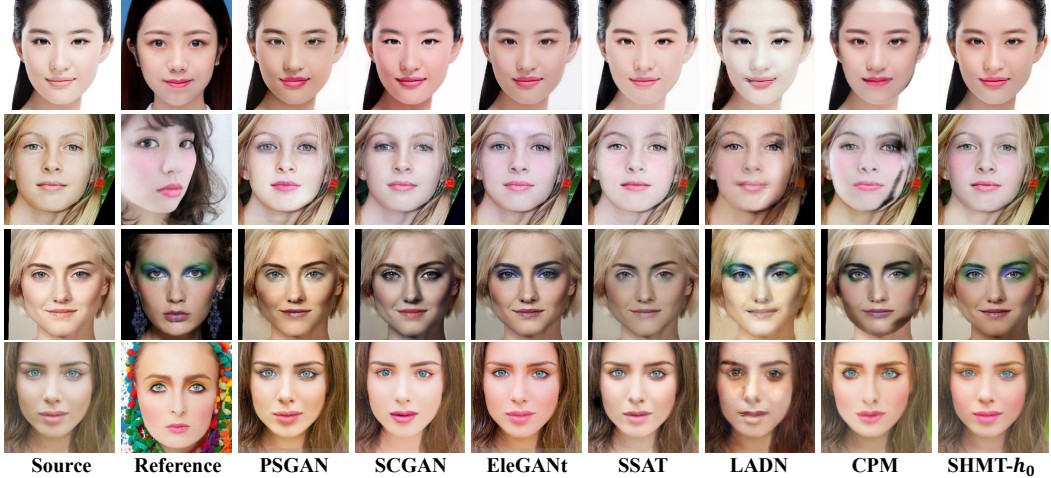

| Source | Reference | PSGAN | SCGAN | EleGANt | SSAT | LADN | CPM | SHMT-$h_0$ |

Figure 4: Qualitative comparison with GAN-based baselines on simple makeup styles.

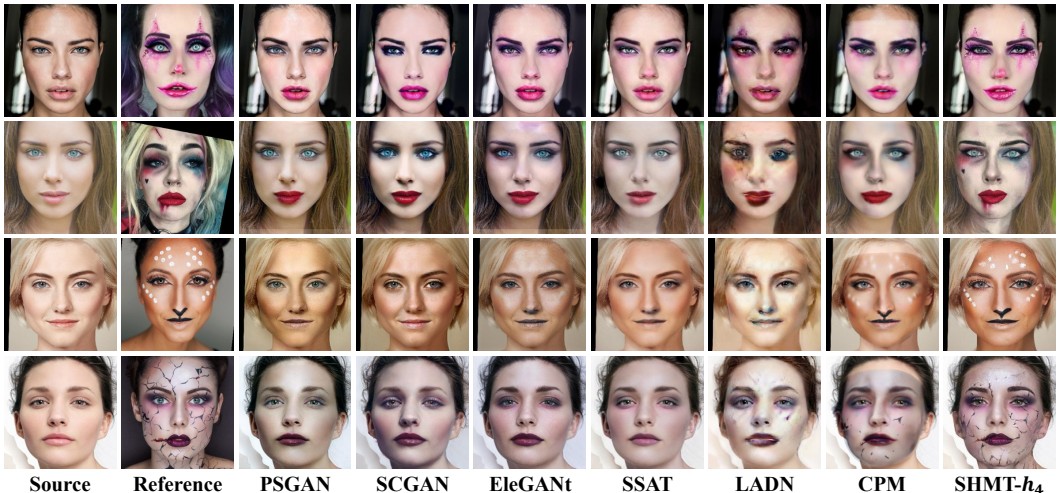

| Source | Reference | PSGAN | SCGAN | EleGANt | SSAT | LADN | CPM | SHMT-$h_4$ |

Figure 5: Qualitative comparison with GAN-based baselines on complex makeup styles.

better. Recently, the work [39] proves that the *CLS* token and the self-similarity of keys (abbreviated as *Key-sim*) in DINO's [50] feature space can represent the appearance and structure of an image, respectively. Inspired by this, we compute the cosine similarity of the *CLS* token between reference images and transferred results to represent the *makeup fidelity*, and the cosine similarity of *Key-sim* between source images and transferred results to reflect the *content preservation*. The higher the *CLS* and *Key-sim*, the better.

**Baselines.** We choose seven state-of-the-art makeup transfer methods as baselines, including PSAGN [17], SCGAN [7], EleGANt [47], SSAT [36], LADN [11], CPM[29] and Stable-Makeup[52]. Among them, only Stable-Makeup is a diffusion-model-based method, while the others are GAN-based methods. And LADN, CPM and Stable-Makeup focus on complex makeup styles. In our experiments, the baseline results are derived from official publicly available code or pre-trained models.

## 4.2 Comparisons

**Qualitative Results.** The qualitative comparison with different GAN-based methods for simple and complex makeup styles are shown in Figure 4 and Figure 5, respectively. Although PSGAN, SCGAN, EleGANt, and SSAT preserve the content of the source image well, they have low fidelity for reference makeup, especially in complex makeup styles. In addition, they have a tendency to modify

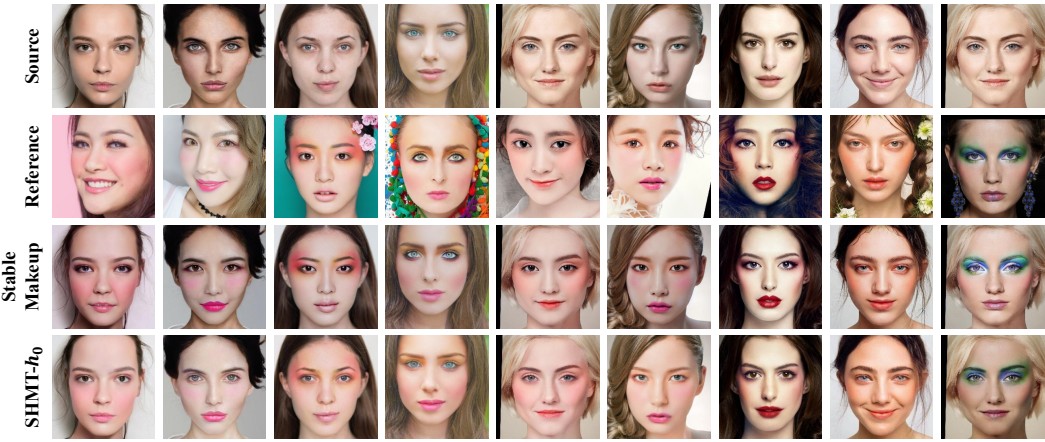

Figure 6: Qualitative comparison with the Stable-Makeup baseline on simple makeup styles.

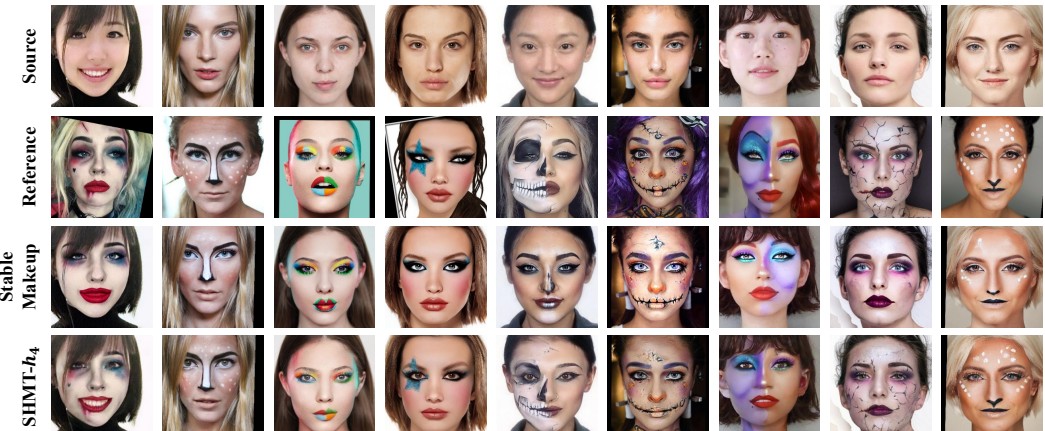

Figure 7: Qualitative comparison with the Stable-Makeup baseline on complex makeup styles.

the background color, e.g., the second and fourth rows of Figure 5. LADN's results are accompanied by a large number of artifacts and content distortions. CPM performs relatively satisfactorily in complex makeup, but still fails to reproduce some high-frequency makeup details. Due to the UV space not including the forehead area, the CPM has a noticeable sense of pasting, and there are some artifacts along the facial contour. The results of diffusion-model-based methods are shown in Figure 6 and Figure 7. For simple makeup styles, the results of Stable-Makeup show a noticeable color shift when compared to the reference makeup styles. In addition, these results tend to alter the content information of the source image, including identity and expression. For complex makeup styles, the results of Stable-Makeup still show a significant loss of high-frequency makeup details. In contrast, our SHMT method can naturally and accurately reproduce various makeup styles on the source face by equipping it with different texture details.

**Quantitative Results.** In our experiments, we randomly selected 1000 source-reference image pairs from the test set of each dataset to calculate each evaluation metric. The quantitative results of the different methods are listed in Table 1. SHMT-$h_0$ equipped with fine texture details $h_0$ obtains the highest value on *Key-sim*, indicating better content preservation. SHMT-$h_4$ equipped with coarse texture details $h_4$ achieves the highest values on *FID*, *CLS*, suggesting greater image realism and makeup fidelity. This also demonstrates that there is a tradeoff between content preservation and makeup fidelity in the makeup transfer task.

**User Study**. We also perform a user study to evaluate the performance of different models. We randomly select 50 pairs of images with different types of makeup styles and generate the transferred

Table 1: Quantitative results of *FID*, *CLS* and *Key-sim* on the MT, Wild-MT and LADN datasets.

| Methods | MT | | | Wild-MT | | | LADN | | |
|---|---|---|---|---|---|---|---|---|---|
| | *FID* | *CLS* | *Key-sim* | *FID* | *CLS* | *Key-sim* | *FID* | *CLS* | *Key-sim* |
| PSGAN [17] | 45.02 | 0.628 | 0.975 | 89.92 | 0.642 | 0.969 | 57.80 | 0.684 | 0.975 |
| SCGAN [7] | 39.20 | 0.636 | 0.965 | 79.54 | 0.660 | **0.976** | 51.39 | 0.685 | 0.973 |
| EleGANt [47] | 54.06 | 0.634 | 0.973 | 86.19 | 0.651 | 0.961 | 61.40 | 0.693 | 0.969 |
| SSAT [36] | 38.01 | 0.645 | 0.975 | 70.53 | 0.667 | 0.973 | 53.84 | 0.692 | 0.976 |
| LADN [11] | 73.91 | 0.620 | 0.917 | 104.91 | 0.634 | 0.914 | 65.87 | 0.688 | 0.930 |
| CPM [29] | 42.76 | 0.652 | 0.951 | 95.61 | 0.661 | 0.924 | 40.57 | 0.729 | 0.954 |
| Stable-Makeup [52] | 33.26 | 0.682 | 0.973 | 64.64 | 0.711 | 0.968 | 37.33 | 0.767 | 0.965 |
| SHMT-$h_0$ | 32.24 | 0.658 | **0.976** | 51.54 | 0.668 | **0.976** | 38.97 | 0.711 | **0.978** |
| SHMT-$h_4$ | **24.93** | **0.715** | 0.953 | **45.02** | **0.719** | 0.954 | **27.01** | **0.786** | 0.958 |

Table 2: The ratio selected as best (%).

| Styles | EleGANt | SSAT | CPM | SHMT |
|---|---|---|---|---|
| Simple | 20.25% | 13.5% | 4.75% | 61.5% |
| Complex | 1.75% | 0% | 11.5% | 86.75% |

Table 3: Quantitative results of IDA on LADN dataset.

| Methods | *FID* | *CLS* | *Key-sim* |
|---|---|---|---|
| SHMT-$h_4$ w/o IDA | 32.42 | 0.753 | 0.960 |
| SHMT-$h_4$ | 27.01 | 0.786 | 0.958 |

results using different methods. Then, 8 participants are asked to choose the most satisfactory result, considering image realism, content preservation, and makeup fidelity. To ensure a fair comparison, the transferred results are displayed simultaneously in a random order. The results of the user study are shown in Table 2.

### 4.3 Ablation Study

**The effectiveness of hierarchical texture details.** Figure 8 (a) illustrates the visual comparison of SHMT with varying texture details in handling both simple and complex makeup styles. SHMT-$h_0$ tends to preserve high-frequency details of the source image, such as subtle expressions, single and double eyelids, eyelashes, and freckles. It is more suitable for simple makeup transfer. On the other hand, SHMT-$h_4$ is more likely to transfer the high-frequency details from the reference image, making it more appropriate for complex makeup transfer. By staggering the timestep and employing a different model to predict the noise, i.e., using SHMT-$h_0$ for [T,t] and SHMT-$h_4$ for (t,0), we can easily achieve a seamless interpolation between the two results.

**The effectiveness of IDA.** To verify the effectiveness of IDA, we remove the additional alignment prediction $z_m''$ and replace $\hat{z}_m$ with $z_m'$ as an injection condition for the denoiser. We retrain the SHMT-$h_4$ model, and the alignment visualizations and transfrred results are shown in Figure 8 (b). As seen, the proposed IDA module effectively corrects alignment errors and enhances makeup fidelity. The quantitative results are displayed in Table 3. In Figure 8 (c), the trend of percentage $w$ over time is investigated. As timestep $t$ becomes smaller, the domain gap between the noisy intermediate result $\hat{I}_t$ and makeup representation $I_m$ gets smaller, and $w$ increases indicating that the model favors more the alignment prediction of these two. In addition, $w$ of SHMT-$h_4$ is significantly higher than that of SHMT-$h_0$ at each timestep $t$, which we attribute to the fact that transfering high-frequency texture details requires more precision in semantic alignment.

**Robustness and Generalization.** Further, as shown in Figure 9 (a), calculating the semantic alignment between the source and reference images makes our model insensitive to age, gender, pose and expression variations. To evaluate the generalization ability, we collect some sketch and anime examples which have a significant domain gap with the training samples and have never been encountered by the model before. The results are displayed in Figure 9 (b).

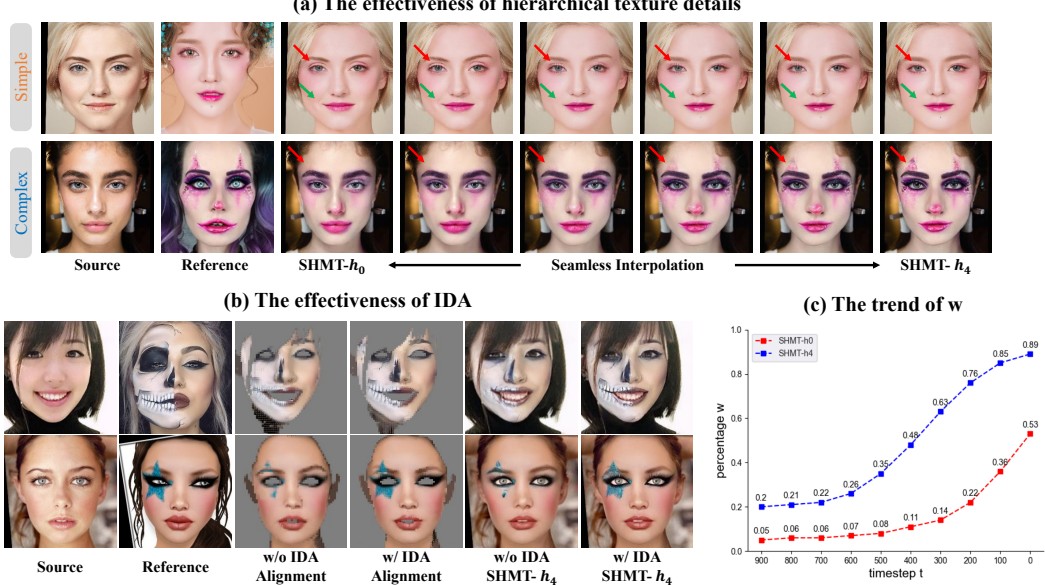

Figure 8: Ablation studies of each proposed module to validate its effectiveness. Zoomed-in view for a better comparison.

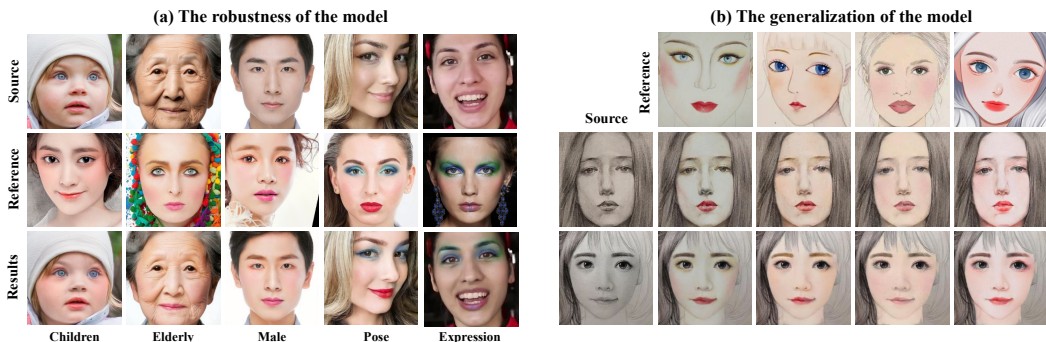

Figure 9: The robustness and generalization ability of the model SHMT-$h_0$ in various scenarios.

## 4.4 Limitations

The limitations of our method can be summarized in two aspects. First, our proposed model relies on the prior knowledge of the pre-trained models (face parsing and 3D reconstruction), and the stability of our model suffers when their output is inaccurate. More results and discussion are provided in the supplementary material. Second, compared to previous GAN-based approaches, our model has more parameters and requires more computational resources during inference, taking several seconds to generate an image. Recent accelerated sampling techniques [25, 53, 23] may be able to alleviate this limitation to some extent.

## 5 Conclusion

In this paper, we propose a Self-supervised Hierarchical Makeup Transfer (SHMT) method. It employs a self-supervised strategy for model training, freeing itself from the misguidance of pseudo-paired data employed by previous methods. Benefiting from hierarchical texture details, SHMT can flexibly control the preservation or discarding of texture details, making it adaptable to various makeup styles. In addition, the proposed IDA module is capable of effectively correcting alignment errors and thus enhancing makeup fidelity. Both quantitative and qualitative analyses have demonstrated the effectiveness of our SHMT method.

## Acknowledgments

This work was in part supported by the National Key Research and Development Program of China (Grant No. 2022ZD0160604), the National Natural Science Foundation of China (Grant No. 62176194), the Young Scientists Fund of the National Natural Science Foundation of China (Grant No. 62306219), the Key Research and Development Program of Hubei Province (Grant No. 2023BAB083), the Project of Sanya Yazhou Bay Science and Technology City (Grant No. SCKJ-JYRC-2022-76, SKJC-2022-PTDX-031), the Project of Sanya Science and Education Innovation Park of Wuhan University of Technology (Grant No. 2021KF0031), Alibaba Group through Alibaba Research Intern Program.

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

# Appendix

## A  The Effectiveness of Hierarchical Texture Details

As the texture details go from fine to coarse, the high-frequency information of the original image provided in the content representation decreases. In order to reconstruct the original image, our model SHMT has to learn more high-frequency details from the makeup representation, thus adapting to more complex makeup styles. To verify this, we train the SHMT model equipped with different texture details separately, and the corresponding results are shown in Figure 10. From $h_0$ to $h_4$, our model gradually shifts from preserving the texture details of the source image to transferring the texture details of the reference image, such as the skeleton makeup style in the third row. The quantitative results of the corresponding models on the LADN dataset are displayed in Table 4. As the metric *CLS* gradually increases, the metric *Key-sim* gradually decreases, which also indicates a trade-off between makeup fidelity and content preservation.

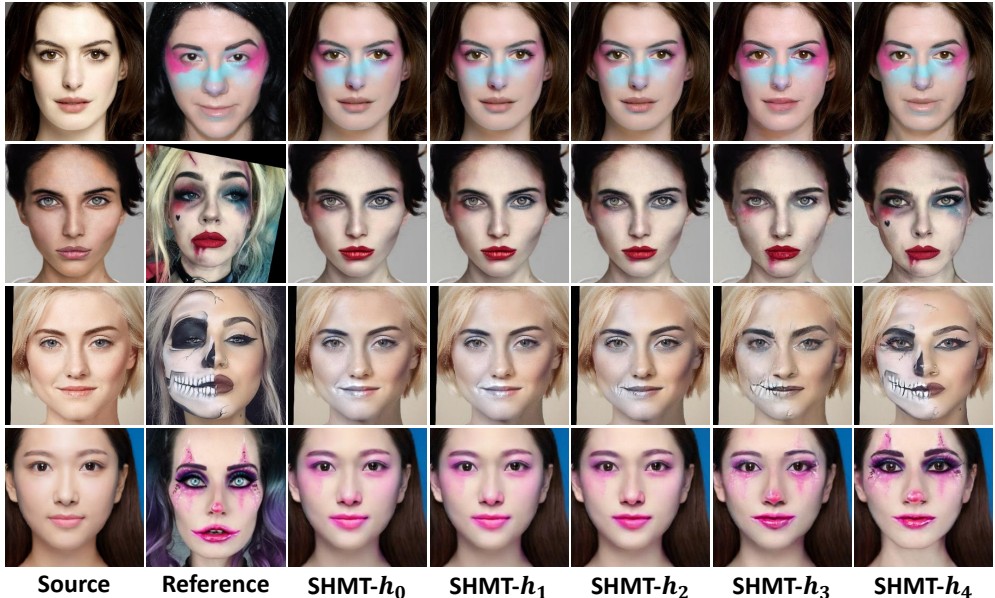

Figure 10: Qualitative results of models equipped with different texture details under complex makeup styles. As the texture goes from fine to coarse, the model gradually tends to transfer high-frequency texture details from the reference images.

Table 4: The quantitative results of our models equipped with different texture details on the LADN dataset.

| Methods | FID | CLS | Key-sim |
|---|---|---|---|
| SHMT-$h_0$ | 38.97 | 0.711 | **0.978** |
| SHMT-$h_1$ | 38.86 | 0.712 | **0.978** |
| SHMT-$h_2$ | 36.35 | 0.726 | 0.973 |
| SHMT-$h_3$ | 33.12 | 0.748 | 0.965 |
| SHMT-$h_4$ | **27.01** | **0.786** | 0.958 |

## B  Comparison with InstantStyle

We also compare the proposed method with the recent style transfer method InstantStyle [42]. The qualitative results are shown in Figure 11. In InstantStyle [42], we fix the text prompt to "a woman, best quality, high quality", the number of samples is set to 4, and other parameter configurations follow the default settings. As seen, InstantStyle [42] captures the color and brushstroke style of the global image, not the makeup style of the face. Additionally, it does not effectively preserve the content information of the source image, indicating that the generalized style transfer method may not be suitable for the specific makeup transfer task.

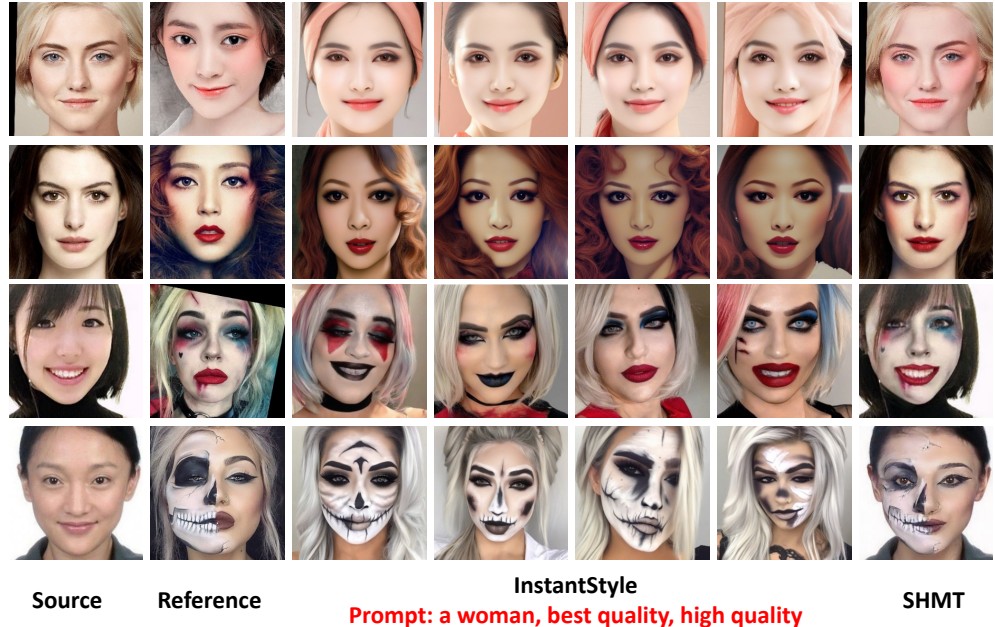

**Source**     **Reference**                     **InstantStyle**                     **SHMT**
                                         Prompt: a woman, best quality, high quality

Figure 11: Qualitative comparison of our method SHMT with the style transfer method InstantStyle.

# C   Makeup Style Control

## C.1   Global Makeup Interpolation

In our proposed method, the makeup information is decoupled from the input images and encoded into the feature maps $\hat{z}_m$. This allows us to interpolate the makeup styles between two different reference faces by linearly fusing their the feature maps $\hat{z}_m$, as follows:

$$\hat{z}_m = (1 - \beta)\hat{z}_{m1} + \beta\hat{z}_{m2}. \tag{5}$$

Here $\hat{z}_{m1}$ and $\hat{z}_{m2}$ are alignment predictions of two different reference images, respectively. By adjusting the value of $\beta$ from 0 to 1, SHMT can generate a series of transferred results. Their makeup styles will gradually change from that of one reference image $y_1$ to that of the other $y_2$. Moreover, by assigning the source image as $y_1$, we can control the degree of makeup transfer for a single reference input $y_2$. The global makeup interpolation results are shown in Figure 12.

## C.2   Local Makeup Interpolation

In SHMT, the alignment prediction $\hat{z}_m$ is deformed through the spatial attention, so that it can be semantically aligned with the source image. Such spatial alignment enables SHMT to implement the makeup interpolation within different local facial areas, which can be formulated as follows:

$$\hat{z}_m = ((1 - \beta)\hat{z}_{m1} + \beta\hat{z}_{m2}) \otimes Mask_{area} + \hat{z}_{m\_self} \otimes (1 - Mask_{area}), \tag{6}$$

where $\otimes$ denotes the Hadamard product and $\hat{z}_{m\_self}$ denotes the alignment prediction by assigning the source image as a reference image. $Mask_{area}$ is a binary mask of the source image $x$, indicating the local areas to be makeup, which can be obtained by face parsing. Figure 13 visualizes the local makeup interpolation results within the areas around the lips and eyes, respectively, i.e., $area \in lip, eye$ for $Mask_{area}$. Similarly, we can also control the local makeup transfer degree of a single reference image by replacing the other reference input with the source image.

## C.3   Preserving Skin Tone

Similar to previous approaches [17, 36, 47, 7, 46, 11, 46, 52], SHMT assumes that the foundations and other cosmetics have already covered the original skin tone. Therefore, the skin color of the reference face is considered as a part of its makeup styles and is faithfully transferred to the final generated result, which may corrupt the skin tone of the source image. To alleviate this problem, we can perform the above-mentioned local makeup

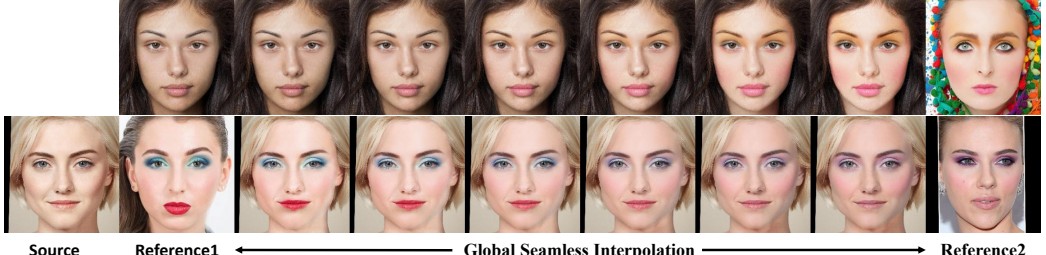

**Source**  **Reference1** ◄————— **Global Seamless Interpolation** —————► **Reference2**

Figure 12: The illustration of global makeup interpolation. The first row is the result of a single reference image, the second row is the result of two reference images. .

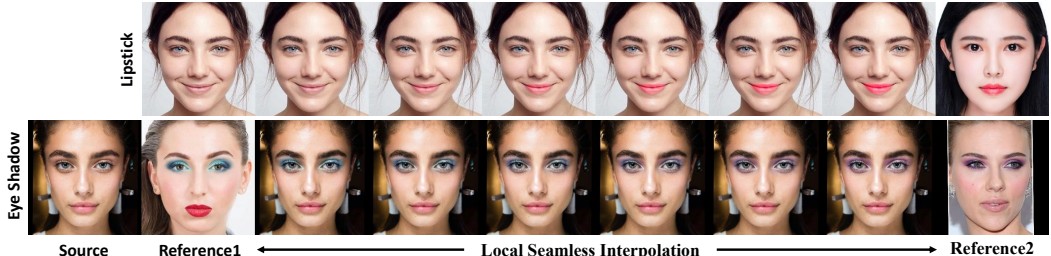

**Source**  **Reference1** ◄————— **Local Seamless Interpolation** —————► **Reference2**

Figure 13: The illustration of local makeup interpolation. The first row is lipstick control, the second row is eye shadow control.

interpolation operation in the face region of the source image to preserve its skin tone. This procedure can be formulated as:

$$\hat{z}_m = ((1 - \beta)\hat{z}_{m\_self} + \beta\hat{z}_{m2}) \otimes Mask_{face} + \hat{z}_{m\_self} \otimes (1 - Mask_{face}). \tag{7}$$

Here, the transferred result realizes the local makeup interpolation between the source image $x$ and the reference image $y_2$ within the face area (excluding the lip and eye areas) in $x$, which is indicated by the mask $Mask_{face}$. The interpolation results are visualized in Figure 14. When $\beta = 0$, the transferred result will not change the skin tone of the source image. And when $\beta = 1$, Equation (7) degenerates to the standard makeup transfer process in SHMT, which will distill the makeup information (including the skin tone) from the reference image to the source image.

### C.4 More Comparison Results

Figure 15 and Figure 16 show more qualitative comparisons between SHMT and state-of-the-art methods on simple and complex makeup styles, respectively.

## D Limitations

Our proposed model relies on the prior knowledge of the pre-trained models (face parsing and 3D reconstruction), and the stability of our model suffers when their output is inaccurate. As shown in Figure 17, the face parsing model often marks high-frequency makeup styles in the forehead area as hair and segments them into the background area, resulting in performance degradation.

## E Social Impact

### E.1 Social Impacts

Facial makeup customization offers an entertaining tool for generating realistic character photos. However, the long-term usage of makeup transfer techniques may increase users' appearance anxiety. And such technology may require access to personal biometric information, such as facial features, which could raise concerns about facial privacy if misused.

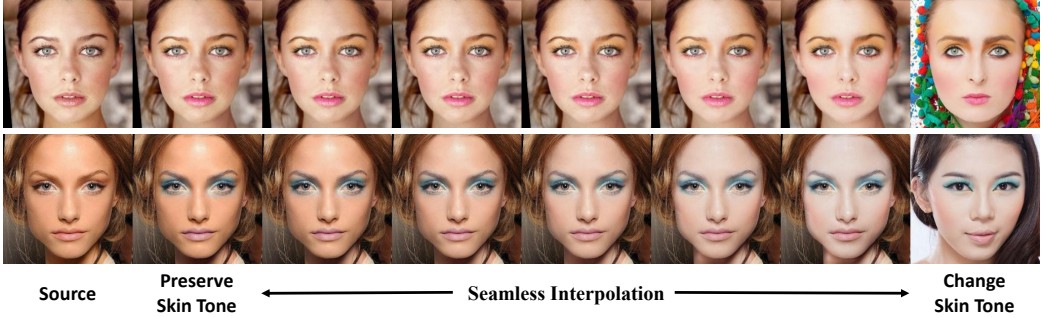

Figure 14: By default, our method transfers makeup to change the skin tone. Optionally, the local makeup transfer operation can preserve the original skin tone, and the local makeup interpolation can smoothly generate intermediate results.

## E.2 Safeguards

We will utilize the following strategies to mitigate negative impacts:

1) We will encrypt or anonymize facial images during transmission and storage, such as using hash values instead of real image data.

2) We will use the Stable diffusion safety checker [3] to conduct security checks on our generated images, so that we can identify and handle Not Safe For Work (NSFW) contents in images.

3) Since our method is working on human faces, we will also employ some deep-fake detection models [1, 6] to filter the results generated by our model.

4) We will ask the users to agree to a license or conform a code of ethics before accessing our model, which requires them to use our model in a more standardized manner.

## E.3 Responsibility to Face Images

The face images in this study are taken from publicly accessible datasets, they're considered less sensitive. Furthermore, our data algorithm is strictly for academic purposes, not commercial use. During the inference stage, the proposed model adjusts only the makeup style without altering the individual's identity, thereby minimizing potential facial privacy concerns.

## E.4 Data Availability

The MT [21], Wild-MT [17] and LADN [11] datasets that used in our experiments have already been released and can be found in the following links:

MT dataset: `https://github.com/wtjiang98/BeautyGAN_pytorch`.

Wild-MT dataset:`https://github.com/wtjiang98/PSGAN`.

LADN dataset: `https://github.com/wangguanzhi/LADN`.

---

[3]https://huggingface.co/CompVis/stable-diffusion-safety-checker

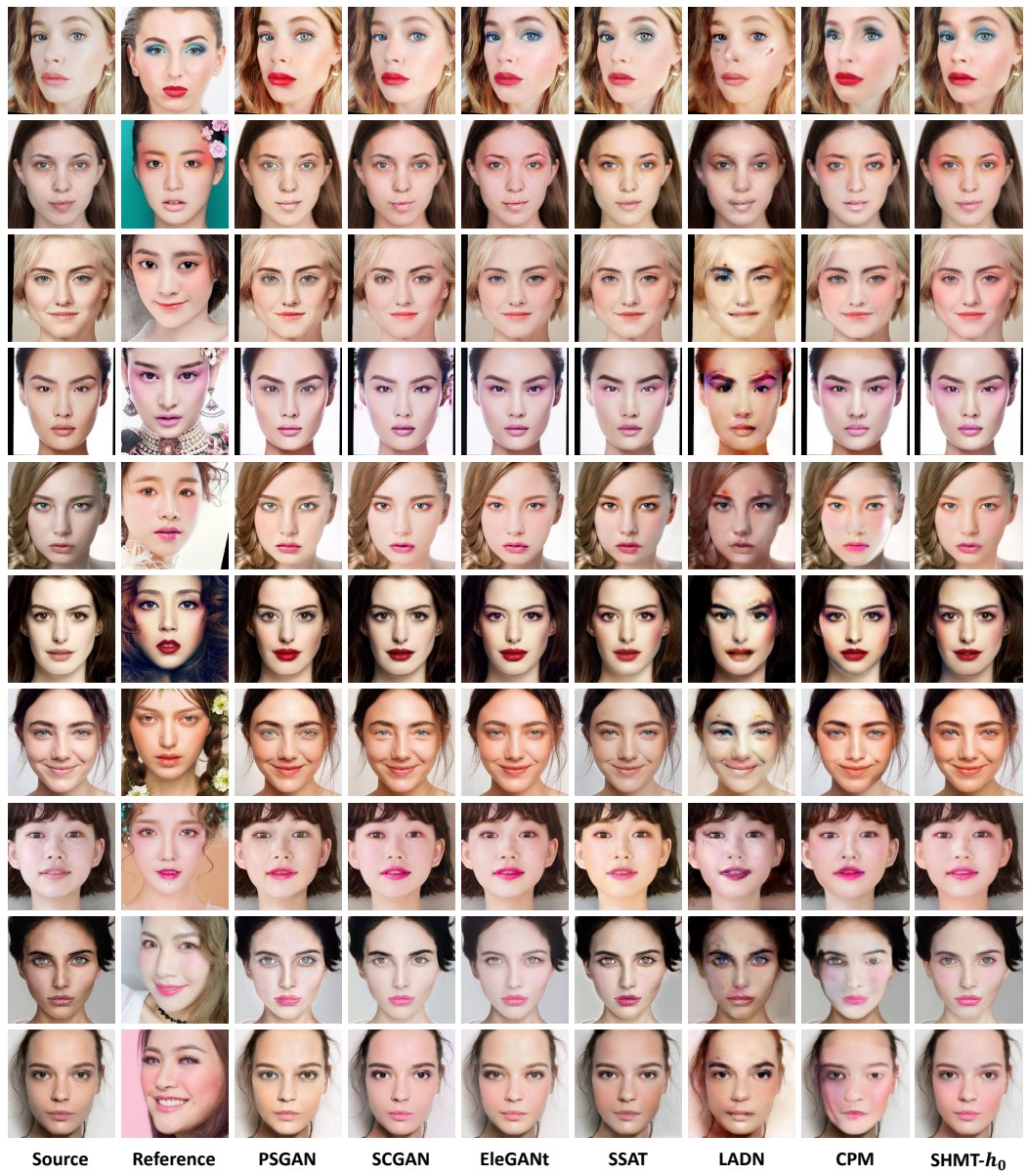

| Source | Reference | PSGAN | SCGAN | EleGANt | SSAT | LADN | CPM | SHMT-$h_0$ |

Figure 15: More qualitative results of different methods in simple makeup styles.

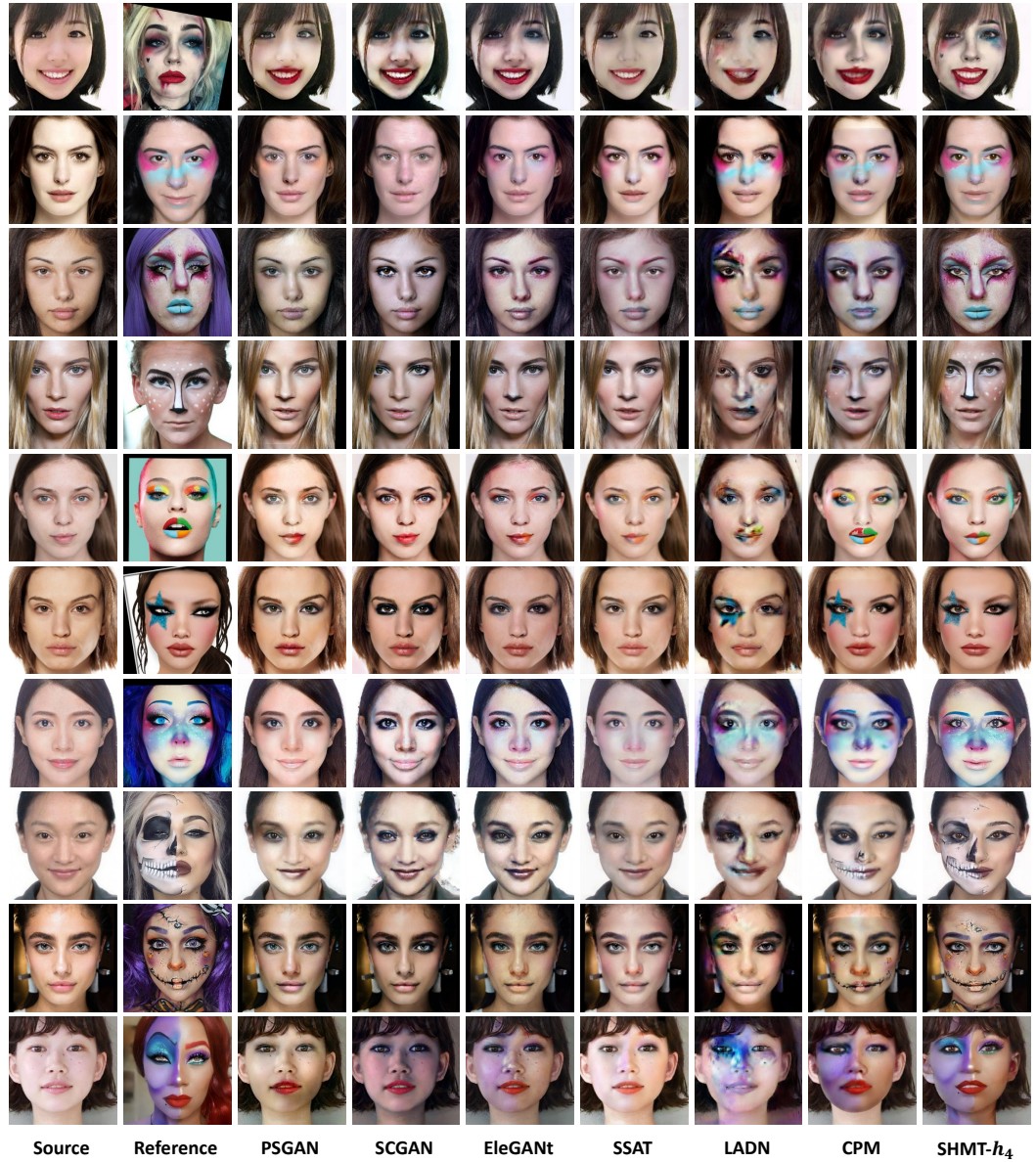

Source    Reference    PSGAN    SCGAN    EleGANt    SSAT    LADN    CPM    SHMT-$h_4$

Figure 16: More qualitative results of different methods in complex makeup styles.

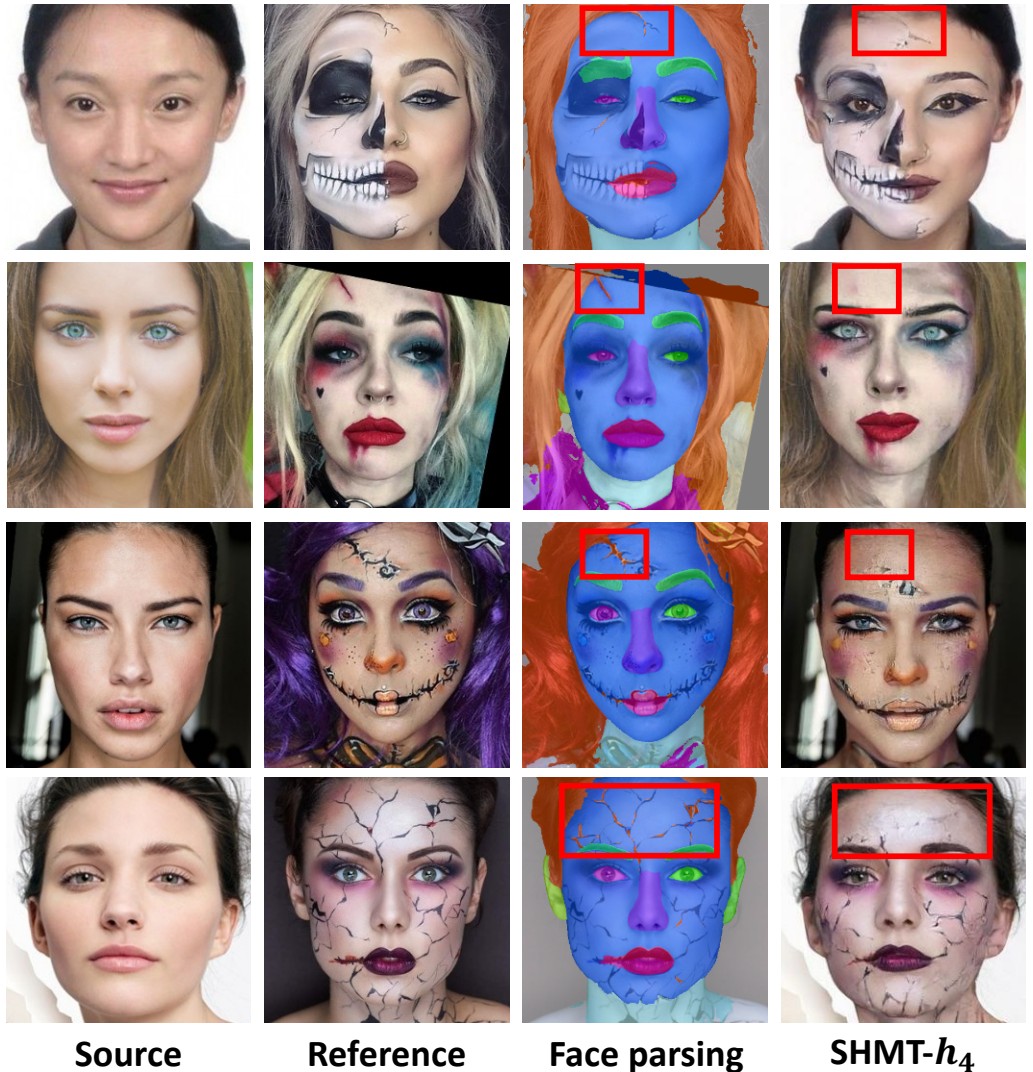

**Source**  **Reference**  **Face parsing**  **SHMT-$h_4$**

Figure 17: Limitations of our approach. The face parsing model often marks high-frequency makeup styles in the forehead area as hair and segments them into the background area, resulting in performance degradation.

