# OpenReview forum: "SHMT: Self-supervised Hierarchical Makeup Transfer via Latent Diffusion Models"
_NeurIPS.cc/2024/Conference — NeurIPS 2024 poster_

### Official Review · Reviewer_dGT5 · 2024-07-09

**Soundness:** 4
**Presentation:** 3
**Contribution:** 2
**Rating:** 7
**Confidence:** 5

**Summary:**

This work introduces a diffusion-based makeup transfer method named Self-supervised Hierarchical Makeup Transfer (SHMT). The proposed method features a network that extracts makeup features from a distorted makeup image and aligns these features with facial features. The facial features consist of two components: one is the noisy generated feature at the current diffusion denoising timestep, and the other is derived from the shape of the source face. These two aligned facial features are then blended and incorporated into the denoising UNet as the makeup condition. Additionally, the face condition allows for fine-grained control of the transfer strength by using pyramid Laplacian features. The coarse Laplacian feature preserves more makeup details, while the fine Laplacian feature retains more details of the face being transferred.

Justification: While using distorted images as a condition to fine-tune a diffusion model for conditioning on a distorted image's appearance is not particularly novel, this work introduces novelty through the control of makeup strength via Laplacian features. Additionally, the experimental section is comprehensive and impressive. For these reasons, I give rating Accept.

**Strengths:**

* The experiments are comprehensive, and the qualitative comparisons between different hyperparameters validate the model’s ability to handle both simple and complex makeup styles.
* The model's ability to generalize to various types of makeup is impressive.

**Weaknesses:**

* It is unclear how the complexity of makeup is judged. Is it determined automatically or by human evaluation? If judged by humans, how is the parameter Laplace feature $h_i$ chosen during training?
* typo: ln 109 Diffusio -> Diffusion

**Questions:**

See weaknesses

**Limitations:**

The limitation has been discussed in the paper.

This study focuses on makeup transfer, necessitating experiments on human faces. Although this approach could potentially generate synthetic faces, the study's subject is makeup transfer rather than face manipulation. Since the human identity is not altered, the reviewer believes this should not raise forensic issues. (This is why I have chosen "no ethics review is needed" for next question.)

---

> ### Author Rebuttal · Authors · 2024-08-06
>
> > 1. It is unclear how the complexity of makeup is judged. Is it determined automatically or by human evaluation? If judged by humans, how is the parameter Laplace feature chosen during training?
>
> Thank you for your question. In our opinion, we believe that each customer who uses a makeup transfer application has his or her own understanding and judgement about the complexity of the target makeup styles in the reference images. Establishing a unified standard for such judgement cannot adaptively meet the needs of all customers. Therefore, our proposed method provides the users with the flexibility to make their own decisions about what makeup information should to be transferred. More specifically, during the training phase, we do not choose Laplace features for each single input image. Instead, we fix a specific texture detail and train the model to transfer the decomposed information at the corresponding frequency level. As a result, we simply train five different models (SHMT-$h_i$, $i=0,1,2,3,4$) by consistently incorporating $h_i$ texture details of all the images, as presented in the Section 1 **The effectiveness of hierarchical texture details** of the supplementary materials. After that, during the inference stage, the users can select and combine different models to generate transferred results depending on their own requirements.
>
> > 2. typo: ln 109 Diffusio -> Diffusion.
>
> Thank you for pointing out this issue. As suggested, we have corrected this typo and also carefully checked the whole paper to avoid such mistakes.

---

### Official Review · Reviewer_izVu · 2024-07-10

**Soundness:** 3
**Presentation:** 3
**Contribution:** 3
**Rating:** 6
**Confidence:** 3

**Summary:**

This paper deals with the problem of makeup style transfer given a certain facial image. Current methods usually use synthesized ground truths to guide the model training, which is sub-optimal. This paper proposes to decompose the hierarchical texture details using a Laplacian pyramid and selectively introduce them to the content representation. Quantitative and qualitative analyses demonstrate the effectiveness of the proposed method.

**Strengths:**

1. The problem of makeup style transfer to a given facial image is an interesting topic.
2. The paper is well written and I appreciate the figures in the paper.
3. The proposed SHMT method can flexibly control the preservation or discarding of hierarchical texture details, which achieves better  quantitative and qualitative results compared to current methods.

**Weaknesses:**

1. In Fig4, it seems the makeup transfer results could be influenced by the features of the reference image. For example, if the given reference image has a darker skin color, then the generated result would have a darker color, which might means the generation results could influenced by some irrelevant features other than the makeup style.
2. The comparison methods seem somehow out-dated to me, they could by replaced by some recent methods based on diffusion models.

**Questions:**

I have some questions about the chosen backbone, which seems a baseline model to me. Why didn't try some stronger diffusion models?

**Limitations:**

The authors discussed the limitations of their paper.

---

> ### Author Rebuttal · Authors · 2024-08-07
>
> > 1. In Fig4, it seems the makeup transfer results could be influenced by the features of the reference image. For example, if the given reference image has a darker skin color, then the generated result would have a darker color, which might means the generation results could influenced by some irrelevant features other than the makeup style.
>
> Thank you for your question. In real life, the heavy use of cosmetics can cover up an individual's original skin tone, especially with some complex makeup styles. To the best of our knowledge, nearly all existing makeup transfer methods, including our approach, are unable to distinguish whether the skin color in the reference image is natural or cosmetically altered. Therefore, these methods typically assume that the makeup process will change the original skin tone and do not consider the skin tone preservation. To address this issue, we provide a solution in Section 3.3 Preserving Skin Tone of the supplementary materials. Specifically, we can utilize local makeup interpolation operations to flexibly control the extent of skin tone preservation, as demonstrated in Figure 5 of the supplementary materials.
>
>  > 2. The comparison methods seem somehow out-dated to me, they could by replaced by some recent methods based on diffusion models.
>
> Thank you for pointing out this issue. To the best of our knowledge, there is only one makeup transfer method based on the diffusion model, namely StableMakeup [1], which is currently preprinted on arXiv. We also cite this reference in our related work section as **''With the help of the unprecedented generative capabilities of both GPT-4V and Stable Diffusion, a concurrent work produces higher quality pseudo-paired data, thereby improving the performance of makeup transfer.''** This method still needs to construct pseudo ground truths (PGTs) for model training, while our proposed method adopts a fully self-supervised training strategy, allowing it to avoid the negative effects of PGTs. Nevertheless, during the peer review process of NeurIPS, the authors of StableMakeup release their source code, so we make additional quantitative comparisons (please see Table 1, Figure 1 and Figure 2 in the Global Author Rebuttal PDF). Moreover, we also compare our method with InstantStyle [2], a diffusion model for style transfer tasks. From both the qualitative and quantitative results, we can find that SHMT consistently outperforms these two state-of-the-art diffusion-model-based approaches, indicating the effectiveness of our method. We will also add these results to the final version of our paper.
>
> [1] Zhang, Yuxuan, et al. Stable-Makeup: When Real-World Makeup Transfer Meets Diffusion Model. arXiv preprint arXiv:2403.07764, 2024.
>
> [2] Wang, Haofan, et al. Instantstyle: Free lunch towards style-preserving in text-to-image generation. arXiv preprint arXiv:2404.02733, 2024.
>
> > 3. I have some questions about the chosen backbone, which seems a baseline model to me. Why didn't try some stronger diffusion models?
>
> Thank you for your question. In this paper, we choose the original LDM as our backbone based on following reasons: 1) We need to train our model from scratch, but our computational resources are limited. Considering training costs and time used, we select to use the LDM model. 2) The major contributions of this paper are to design a fully self-supervised diffusion-model-based makeup transfer framework and incorporate the Laplacian pyramid to hierarchically characterize the texture information. Pursuing better performance **by replacing more powerful backbones** is not our main purpose. And moreover, the experiments in our paper demonstrate that using LDM as the backbone can already lead to superior results than previous works. Nevertheless, we believe that employing a stronger diffusion backbone like SD v1.5 or SDXL v1.0 can further empower our SHMT framework and we leave exploring these models in our future works. We appreciate your valuable suggestion.

---

> > ### Comment · Reviewer_izVu · 2024-08-08
> >
> > Most of my concerns are addressed. I still don't totally agree with "Pursuing better performance by replacing more powerful backbones is not our main purpose", but I can understand it. I have one concern left. Though I think ethical concern is not a main issue about this paper, if the authors using widely used datasets in existing makeup transfer to train their model, could the authors upload their from-scratch training code in an anonymous github link. I would be glad to raise my score if the authors could open-source the training code for the development of this field.

---

> ### Author Response · Authors · 2024-08-08
> **Open-source the code of our SHMT method**
>
> Thank you for your reply. As suggested, we have released the training and inference code of our SHMT method in this anonymous github link: https://anonymous.4open.science/r/SHMT-8754/README.md. Following the steps in README, the users can train their own model from scratch and evaluate it. After the anonymous peer review process, we will also open source an official version of the code and the checkpoint weights of our trained models. We hope that open-sourcing our code will alleviate your concerns and contribute to the development of the related research fields. If you can raise your score after checking our code, we would be very appreciated!

---

> ### Comment · Reviewer_izVu · 2024-08-09
>
> Thanks the authors for providing their training code. Nowadays, only open-sourcing inference code is not healthy for the development of the AIGC field. Given the provided code and other reviewers' comments, though the paper is trying to use LDM to solve makeup transfer which uses many existing techniques (limited novelty), I do appreciate the paper writing and the plotted figures, thus I believe the paper is marginally above the acceptance bar of this conference. I have updated my final rating to 6.

---

### Official Review · Reviewer_JmHE · 2024-07-11

**Soundness:** 2
**Presentation:** 3
**Contribution:** 2
**Rating:** 3
**Confidence:** 4

**Summary:**

The paper proposes a method for improving the natural look resulting after makeup transfer. The method is derived from  Latent Diffusion Mode and is "destroys" the content to distillate the makeup, that is further difused. The method is evaluated on MT, Wild-MT, LADN datasets.

**Strengths:**

1. Visual appeal of the results: indeed the assumed purpose (i.e. more natural look) has been, in my view, achieved.
2. The method is tuned for makeup as prior information about the face shape is incorporated.
3. Objective evaluation  show method superiority. The evaluation is convincing.

**Weaknesses:**

1. Ethical concerns (please see below)
2. The technical innovation is limited as mainly there are several prior works put together (although in a non-obvious way) and applied to a new theme (i.e. makeup transfer)
3. The theme of the paper:
 - all the papers used in comparison have been published at ACM-MM, CVPR, ICCV. They are strong recent papers, there is nothing wrong with comparison. Yet,
- NIPS is more machine learning oriented. Strictly from a machine learning point of view, the paper is not very interesting, as all the models are known. The particular novelty is by incorporating face shape
 - thus, the paper is more suitable to ACM-MM, CVPR, Face and Gesture, etc.

**Questions:**

Beyond the weaknesses, I do not have questions. I see the paper fair: it explains what has been done, it improves over prior art, the results look good.

In the rebuttal it would be nice to address, as much as possible, the ethical concerns. Some  emphasis could be on which "checks" might be useful.

However, I still have concerns regarding the fit of the theme with NIPS. Unfortunately, the theme of the paper cannot be changed in the rebuttal.

**Limitations:**

Unfortunately, the ethical impact has not been addressed properly. The approach is Section 6 of the supplementary material:
1. Potential risk: it has been accepted that "Facial customization for makeup transfer offers an entertaining tool for generating realistic character photos. However, if misused, it could potentially produce false information." I agree with that : the risk has been acknowledged.

2. Checks. It has been pointed that "Moving forward, we should implement checks on generated photos to minimize any negative impacts." In only partially agree with that. The major difference in opinion is that, in my view, those "checks" should have been listed, or at least hinted and be part of the method (main paper). Now, having only a vague promise, it is too weak.

3. Data license
- It has been argued that "The human images collected in this study come from publicly available sources with open copyrights. As most images feature public figures, they’re considered less sensitive." Further it has been pointed to "Licenses for Datasets" (supplementary - l 70). Yet at the respective URL locations there are licenses for software and not for images. In one case there is  accepted that no license is available.
- it has been said that "Furthermore, our data algorithm is strictly for academic purposes, not commercial use". Yet in the main paper at l. 544 it has been said "The release of the code is subject to the company’s permission, and we will do our best to release the code and trained models as soon as possible.". The problem here is the contradiction between "academic" and "our company".

In summary:
Pro:
  -  I believe that paper acknowledges the risk
Con:
 - it does insufficiently to address them, delivering only a vague promise and to ensure that data (images) are properly used.

---

> ### Author Rebuttal · Authors · 2024-08-07
>
> > 1. The technical innovation is limited as mainly there are several prior works put together (although in a non-obvious way) and applied to a new theme (i.e. makeup transfer).
> 2. Strictly from a machine learning point of view, the paper is not very interesting, as all the models are known.
>
> Thank you for your question. But we cannot agree with your comments saying that “The technical innovation is limited as mainly there are several prior works put together.” Specifically, our method develops a new self-supervised learning strategy for training the diffusion-model-based makeup transfer framework without generating pseudo ground truths as previous works. **This strategy not only makes the semantic alignment between the makeup image $I_m$ and the intermediate noisy image $\hat{I_t}$, but also semantically align $I_m$ with the content details $I_{3d}$ and $h_{i}$.** In this way, the makeup styles in reference images and the content information in source images can be fully integrated into each diffusion denoising step, so that the generated performance can be improved. Moreover, we also associate different frequency components (decomposed by a Laplacian pyramid) with different complexity levels of makeup styles, and **for the first time unify the modeling of both simple and complex makeup style information in a single framework, which is a novel technique that has not been explored in previous works.** This novelty is also approved by Reviewer dGT5 “this work introduces novelty through the control of makeup strength via Laplacian features.” Based on the above analysis, we believe that our proposed method exhibits sufficient technical innovation.
>
> We also cannot agree with the comments that “the paper is not very interesting, as all the models are known.” In our opinion, it would be unreasonable to give our paper a negative rating for this reason. Although the basic network components of our framework have been proposed in previous works, we utilize them to **solve a novel academic problem, i.e., how to design a unified framework that can preserve the source content details for simple makeup styles and also discard those details for complex makeup transfer.** Such motivation has been clearly stated in our introduction section. To solve this problem, we explore to employ the diffusion model to establish our SHMT framework (existing methods are built based on GAN), and integrate the novel designs mentioned in the above paragraph. Therefore, we believe that our paper has proposed a novel method to solve a new problem, which will make a specific contribution to relevant research community.
>
> > 3. The theme of the paper: All the papers used in comparison have been published at ACM-MM, CVPR, ICCV. They are strong recent papers, there is nothing wrong with comparison. Yet, NIPS is more machine learning oriented. Strictly from a machine learning point of view, the paper is not very interesting, as all the models are known. The particular novelty is by incorporating face shape, thus, the paper is more suitable to ACM-MM, CVPR, Face and Gesture, etc.
>
> Thank you for your comments. We quote the call for paper of NeurIPS 2024 from its official website [https://neurips.cc/Conferences/2024/CallForPapers] as follow:
>
> > “The Thirty-Eighth Annual Conference on Neural Information Processing Systems (NeurIPS 2024) is an interdisciplinary conference that brings together researchers in machine learning, neuroscience, statistics, optimization, **computer vision**, natural language processing, life sciences, natural sciences, social sciences, and other adjacent fields. We invite submissions presenting new and original research on topics including but not limited to the following:
> >
> >•	**Applications (e.g., vision, language, speech and audio, Creative AI)**
> >
> >•	Deep learning (e.g., architectures, **generative models**, optimization for deep networks, foundation models, LLMs) ”
>
> The makeup transfer task and our paper clearly align with the topics of **Applications (vision)** and **Deep learning (generative models)**. Moreover, our novelty is not only “incorporating face shape”. As mentioned in the first question, we deign a new self-supervised training strategy based on double semantic alignment to avoid the negative effects of the pseudo ground truths. In addition, we also integrate the modeling of both simple and complex makeup into a unified framework, which enables the users to control the makeup strength more flexible. Finally, extensive quantitative and qualitative analyses demonstrate the effectiveness of our method. Therefore, we think that our paper is fit the theme with NeurIPS 2024.

---

> > ### Author Response · Authors · 2024-08-07
> > **The response of the ethical concerns**
> >
> > Please see the Global Author Rebuttal for the response of the ethical concerns.

---

> > > ### Comment · Reviewer_JmHE · 2024-08-12
> > > **Thoughts on ethical concerns**
> > >
> > > Again let me thank the authors for their effort in addressing the concerns!
> > >
> > > In summary I am keeping my opinion that ethical concerns are not addressed properly.
> > >
> > > The use of make-up transfer can be use in denigrative ways, for instance, transferring features that are considered shaming in the respective society. For instance, in a gender conservative society, creating make-up with opposite gender features. The checks suggested in the response point to references dealing with identity tracking, and to be honestly, I cannot figure to what (e.g. stable diffusion checker).
> > > The " license or conform a code of ethics" that will be imposed to uses is, also, un-detailed. The fact that databases are public and popular, in my view, does not solve the license and user agreement problem. Making the code public does not help with ethical problems. As ethical reviewer pointed, it is expected to keep track and secure carefully the use.
> > >
> > > Best regards.

---

> > ### Comment · Reviewer_JmHE · 2024-08-12
> > **Contribution to machine learning**
> >
> > Thank you for your detailed and thoughtful answer!
> >
> > Unfortunately, In my view, the contribution to machine learning is not evaluated properly. The evaluation focuses only on the make-up transfer. To stand alone, the contribution should have been evaluated on several different problems. As it is, the paper focuses on the make-up transfer and to address noted limitations, it improves on (machine learning) algorithmic part. Yet the paper does not prove that the improvement goes beyond the approached theme.
> >
> > I am keeping my opinion about poor fit with NIPS. Yet this easy to judge by area chairs and program chairs. If this would be ACM-MM, the paper would be fine.
> >
> > Best regards!

---

### Author Rebuttal · Authors · 2024-08-07

Here, we attempt to solve the ethical concerns of Reviewer JmHE as follows:
> 1. Ethical concerns: Potential risk: it has been accepted that "Facial customization for makeup transfer offers an entertaining tool for generating realistic character photos. However, if misused, it could potentially produce false information." I agree with that : the risk has been acknowledged.

Thank you for your question. Our proposed SHMT model mainly focuses on transferring the makeup styles from the reference image to the source face, which are two user-specified inputs. During the transfer procedure, the identity information of both source and reference faces will be maximumly preserved. This is also supported by reviewer dGT5 that “This study focuses on makeup transfer, necessitating experiments on human faces. Although this approach could potentially generate synthetic faces, the study's subject is makeup transfer rather than face manipulation. Since the human identity is not altered, the reviewer believes this should not raise forensic issues.” And moreover, our SHMT trains the LDM from scratch rather than fine-tuning a pre-trained model, which will prevent the obtained model from memorizing the characteristics of particular individuals and also avoid to generate a person-specific image without providing source image. This further circumvents potential ethical risks.

> 2. Ethical concerns: Checks. It has been pointed that "Moving forward, we should implement checks on generated photos to minimize any negative impacts." In only partially agree with that. The major difference in opinion is that, in my view, those "checks" should have been listed, or at least hinted and be part of the method (main paper). Now, having only a vague promise, it is too weak.

Thank you for pointing out this issue. We list the specific checking operations as follows:

1. We will utilize the Stable diffusion safety checker [https://huggingface.co/CompVis/stable-diffusion-safety-checker] to conduct security checks on our generated images, so that we can identify and handle Not Safe For Work (NSFW) contents in images.

2. Since our method is working on human faces, we will also employ some deep-fake detection models [1][2] to filter the results generated by our model.

3. We will ask the users to agree to a license or conform a code of ethics before accessing our model, which requires them to use our model more standardizedly.

[1] Aghasanli, Agil, Dmitry Kangin, and Plamen Angelov. Interpretable-through-prototypes deepfake detection for diffusion models. Proceedings of the IEEE/CVF international conference on computer vision. 2023.

[2] Corvi, Riccardo, et al. On the detection of synthetic images generated by diffusion models. IEEE International Conference on Acoustics, Speech and Signal Processing (ICASSP), 2023.

We will clearly list the above checking operations in the final version of our paper, and we hope these measures can alleviate your concerns about the checking procedure.

> 3. Ethical concerns: Data license: It has been argued that "The human images collected in this study come from publicly available sources with open copyrights. As most images feature public figures, they’re considered less sensitive." Further it has been pointed to "Licenses for Datasets" (supplementary-l 70). Yet at the respective URL locations there are licenses for software and not for images. In one case there is accepted that no license is available.

Thank you for pointing out this issue. We have checked the URL in our paper, and indeed only found the licenses for the software. We will correct these errors in the final version of our paper. Nevertheless, the datasets used in our paper is downloaded from the link provided by their owners (we have properly cited the related original papers and provided the URL locations of the corresponding projects). And moreover, the MT, Wild-MT and LADN datasets are three popular publicly available datasets that have been widely used in existing makeup transfer or privacy protection approaches [1][2][3]. Therefore, we believe that the usage of these datasets has not violated the license of existing assets for NeurIPS 2024.

[1] Yang, Chenyu, et al. Elegant: Exquisite and locally editable gan for makeup transfer. European Conference on Computer Vision. Cham: Springer Nature Switzerland, 2022.

[2] Shamshad, Fahad, Muzammal Naseer, and Karthik Nandakumar. Clip2protect: Protecting facial privacy using text-guided makeup via adversarial latent search. Proceedings of the IEEE/CVF Conference on Computer Vision and Pattern Recognition, 2023.

[3] Sun, Zhaoyang, Yaxiong Chen, and Shengwu Xiong. Ssat: A symmetric semantic-aware transformer network for makeup transfer and removal. Proceedings of the AAAI Conference on artificial intelligence, 2022.

> 4. Ethical concerns: it has been said that "Furthermore, our data algorithm is strictly for academic purposes, not commercial use". Yet in the main paper at l. 544 it has been said "The release of the code is subject to the company’s permission, and we will do our best to release the code and trained models as soon as possible.". The problem here is the contradiction between "academic" and "our company".

Thank you for pointing out this. This paper is an outcome of a collaborative project between our university and a company, which is a fully academic research project rather than a business one. Therefore, the algorithm designed in our paper is strictly for academic purposes, not for commercial use. However, the release of the source code requires the agreement of both parties, so we said that “The release of the code is subject to the company’s permission”. We promise that we will release the code and the model once this paper is accepted.

---

### Decision · Program_Chairs · 2024-09-25

**Decision:**

Accept (poster)

**Comment:**

The technical part and presentation of this paper have been accepted by three reviewers. The paper is well written and presented with interesting and appealing figures. The paper proposes a self-supervised learning method for facial makeup transfer. The proposed method is reasonable and effective, as agreed by reviewers. Authors committed to open-source their code, including training code. The only concerns that have not been well agreed between reviewers and authors are three folds:

(1) Novelty. Reviewers considered that this paper did not bring big innovations to the field, as most of the components are known techniques. However, reviewers agreed that combining the different components to address the makeup transfer problem is not trivial, and particularly the new way of preserving face shape and using the Laplacian pyramid for fine-grained texture transfer. Authors proved the effectiveness of the proposed framework, and showed different configuring ways of the proposed method, as well as the ablation of different components. Overall, AC thinks that the new design is novel enough and especially effective for the focused problem.

(2) Fitness to NeurIPS. One reviewer raised that the paper is not suitable for publication in NeurIPS, mainly due to the lack of machine learning content. After reading the paper, AC regards that though no theoretical machine learning method and proof are presented in this paper, the proposed self-supervised learning method in addressing makeup transfer is still interesting, and presented with clear equations. It is effective in addressing the issue of no paired real-world ground-truth data is available for training. Besides, computer vision in NeurIPS also accepts applied techniques with good performance and deep insights. It's good to see that the paper presents many good insights of the proposed method, including how to control the model behaviours, and what the advantages and limitations are. All the aboves would benefit the computer vision community and perhaps the machine learning community as well in addressing how to design an effective self-supervised learning method for the makeup transfer task where there is no groundtruth, and potentially beyond this tailored task in inspiring other similar learning scenarios.

(3) Ethics issues. The major disagreement among reviewers is the ethics review. Two reviewers thought that no ethics concern existed, while the other reviewer thought the ethics concern was serious in this paper. Additionally, two ethics reviewers helped in providing their expert views of ethics concerns in this paper. They thought that the ethics concerns can be addressed, and provided with useful suggestions in dealing with data usage and the negative impacts regarding face generation. Authors responded positively; they confirmed some actions that had been taken, and some future actions that would be implemented. AC thinks that these positive steps are useful and should be enough to address the ethics concerns.

Though many previously accepted papers did not perform such steps in protecting privacy and prevent negative social impacts and so it appears that it is a bit too strict for the regulation of this paper (from authors' point of view), AC agrees that to make our community healthy and push forward proper use of AI techniques, we should step by step require authors to pay more attention to these ethics concerns. I think authors of this paper have already understood this and agreed to implement the suggested actions. However, this AC would also suggest reviewers and our organizing committees do not over-emphasize or over-interpret the ethics guidelines; otherwise, it would also be a potential harm to the research community in conducting their research and get papers accepted. For example, for dataset usage, we may need to enforce the regulation step by step but not in a hurry. For historical reasons, many existing datasets have not been developed in an ethical way but there have already been many results published on those datasets while replacements may not be available. In this regard, authors have to compare their results to published results. As long as those popular datasets have not been officially withdrawn or be banned by the organizing committee, this AC thinks that authors are still permitted to use those popular datasets at the moment, but certainly, in a guide of knowing the risks and positively looking for replacements. A counter example is the Duke-MTMC dataset, which has already been officially retracted and both NeurIPS and CVPR conferences explicitly require removing this dataset from the submitted papers or otherwise they cannot be accepted; besides, there are many replacements to this dataset.

To conclude, AC regards that this paper presents good value to the community and thus is happy to accept it for publication. Authors are required to include necessary discussions in the rebuttal to the camera-ready version of this paper, implement suggested actions to address the ethics concerns, and the most important, release the source code including the training code to the public as committed, so as to allow other researchers to reimplement the proposed method, verify the results, and implement further improvements.